# Adiabatic Chiral Magnetic Effect in Weyl semimetal wires

Artem Ivashko[1,2], Vadim Cheianov[1,2*]

**1** Instituut-Lorentz for Theoretical Physics, Universiteit Leiden, Niels Bohrweg 2, 2300 RA Leiden, The Netherlands
**2** Delta Institute for Theoretical Physics, Science Park 904, 1090 GL Amsterdam, The Netherlands
* cheianov@lorentz.leidenuniv.nl

December 18, 2017

## Abstract

**The Chiral Magnetic Effect (CME) is a phenomenon by which an electric current develops in the direction of a magnetic field applied to a material. Recent theoretical research suggests that the CME can be observed in thermal equilibrium provided that the magnetic field oscillates with finite frequency $\omega$. Moreover, under certain conditions the amplitude of the electric current does not vanish in the adiabatic $\omega \to 0$ limit, which we call here the adiabatic CME. In this work, we consider an adiabatic CME in a bounded sample. We demonstrate that the presence of the boundary significantly changes the nature of the effect. Using linear response theory, we derive a simple formula (conjectured earlier in [1]) that enables us to describe the CME in a bounded setting based on the knowledge of the energy levels alone. We use this formula to investigate a particularly interesting example of the CME completely defined by the boundary effects.**

## 1 Introduction

Recent years have witnessed an explosion of interest in materials characterised by a topologically non-trivial elementary excitation manifolds. Such interest is largely due to the deep connection that exists between topological characteristics of the elementary excitations and certain non-dissipative transport phenomena. One such phenomenon is the hypothetical Chiral Magnetic Effect (CME), by which an electrical current develops in response to an external *magnetic field* $\boldsymbol{B}$, such that the electrical current density has the form

$$\boldsymbol{j} = \frac{e^2}{h^2 c} \hat{\varkappa} \boldsymbol{B}. \tag{1}$$

Here $\hat{\varkappa}$ is some tensor that reflects the state of the material. The effect was originally predicted in 1980 [2] for non-equilibrium ultra-relativistic plasmas. Later on it was discussed in the context of heavy-ion collisions [3–5], the large-scale dynamics of the early Universe [6–8], magnetohydrodynamics of relativistic plasmas in general [9–11], and superfluid $^3$He-A [12]. In all these systems, the CME arises from the presence of the electro-magnetically charged Weyl fermions in the single-particle excitation spectrum. A Weyl fermion is characterised by a good quantum number called the chirality, which can be either right or left. Each chirality

on its own is described by topologically non-trivial field theory in the following two senses (1) The Berry curvature in the single-particle momentum space has the form of the magnetic field of a monopole of charge +1 for the right-handed and -1 for the left-handed species, and (2) The theory is anomalous, that is the partial electric current of the given chirality is not conserved at quantum level in the presence of an electro-magnetic field. These two topological aspects are deeply related to each other [13], and can be seen as giving rise to the CME [14], which develops when the right- and the left-chiral species are populated with different chemical potentials, $\mu_{\mathrm{R}}$ and $\mu_{\mathrm{L}}$. In the simplest case of Weyl fermions with no additional internal degrees of freedom, the current is parallel to the magnetic field, and $(\hat{\varkappa})_{ij} = \delta_{ij}(\mu_{\mathrm{R}} - \mu_{\mathrm{L}})$.

Recently, Weyl Semi-Metals (WSMs) have attracted a great deal of attention (for recent reviews, see [15, 16]). In these crystalline materials, the electronic Fermi surface consists of several disjoint pockets each surrounding a Weyl node, that is a point in the reciprocal space where the Berry curvature is singular. The effective Hamiltonian in the vicinity of each Weyl node can be brought to the canonical Weyl form by a linear (generally non-orthogonal) coordinate transformation. Thus, the geometry of the $U(1)$ principle bundle of the Bloch wave functions in the vicinity of the Weyl node is identical to that of a Weyl fermion of a given left or right chirality. Each Weyl node can therefore be assigned a chirality left or right. Weyl materials seem to be natural candidate systems for the observation of the CME in laboratory conditions.[1] The most straightforward way to engineer conditions for the CME in a Weyl material would be to apply a constant magnetic field to a sample having an imbalance between the chemical potentials of the left-handed and the right-handed Fermi pockets, $\mu_{\mathrm{L}} \neq \mu_{\mathrm{R}}$. Such an experiment would be of considerable interest due to the hypothetical possibility of a novel type of magnetic instability [19, 20] which has interesting cosmological implications [6, 8]. It is to be noted, however, that in a realistic solid-state system chirality flipping scattering is likely to be quite efficient therefore any such experiment would require constant external driving in order to maintain the chemical potential imbalance. The latter can be implemented, for example, by applying an electric field $\boldsymbol{E}$ parallel to the magnetic field, $\boldsymbol{E} \parallel \boldsymbol{B}$. In such a case, the mechanism responsible for pumping is the chiral anomaly mentioned above [21, 22], and it is actually believed to be the primary cause of the negative longitudinal magnetoresistance that is observed in transport experiments on WSMs [23–27].

Another relatively simple way to drive the system out of equilibrium is to make the magnetic field itself time-dependent, $\boldsymbol{B}(t, \boldsymbol{x}) = \boldsymbol{B}(\boldsymbol{x})\cos(\omega t)$. (Here we have introduced both time and *spatial* dependence of the magnetic field.) In this paper, we will focus exclusively on this type of driving. The bulk CME arising from such a perturbation has been theoretically investigated in recent literature [17, 18, 28–31], basing either on the Kubo formula of the linear response theory or on the semiclassical kinetic equation with the Berry curvature term [32, 33]. These studies converge in their conclusion that a periodically oscillating magnetic field should result in a CME in the form (1) in which the constant $\varkappa$ is proportional to the energy separation $\Delta\varepsilon_0$ between the left-handed and the right-handed Weyl nodes.

Another important finding of the studies is that the current is non-vanishing only in a particular AC limit, $\omega \gg v_{\mathrm{F}}k$, where the coefficient of proportionality between $\varkappa$ and $\Delta\varepsilon_0$ is *universal*. ($\omega$ is the frequency of the field, $k$ is the wavenumber of the field, $v_{\mathrm{F}}$ is a typical velocity of quasiparticles.) On the other hand, in the opposite limit, $\omega \ll v_{\mathrm{F}}k$, the coefficient vanishes.[2] Thus, if we want to probe the universal behaviour in the low-frequency limit $\omega \to 0$,

---

[1]Condensed matter systems offer a broader range of possibilities for the realisation of the CME than Weyl semimetals. Other examples are given in [17, 18].

[2]Note that in order to distinguish the AC limit of the current response from the CME that is driven by the

we are forced to consider delocalized configurations of magnetic field. In other words, the DC response strongly *non-local*. Once the radius of the region of support of the magnetic field exceeds the sample size, it raises the issue of the boundary conditions. The question is: does the $\omega \to 0$ limit of CME survive in this case?

Recent work [1] uses a particular model of a Weyl semimetal to demonstrate that the effect *does* survive in a bounded sample. (See similar conclusion in [18].) Moreover, it was argued in [1] that the surface states contribute significantly: their share in the total current does not decrease in the limit of large sample sizes. These important findings of [1] were obtained by combining numerical simulations of the spectrum of a bounded sample of a Weyl material with heuristic formula for the CME, inspired by the Landauer-Büttiker approach. Here, we derive the generalized Landauer-Büttiker formula of [1] from the first-principle quantum-mechanical approach to linear response (Kubo formula) for *bounded* systems. We also confirm the findings of [1] regarding the well-defined limit $\omega \to 0$ for a bounded sample.

Before we proceed to the complete quantum-mechanical framework describing the CME in a bounded system, let us briefly discuss the fundamentally non-local nature of Eq. (1) and the implications of this non-locality for systems with boundaries. For this, we will use the semiclassical approach of [17,31], which will also enable us to illustrate one of the non-trivial effects of the boundary. In an infinite system, the semiclassical expression for the current density is

$$\boldsymbol{j} = \frac{e}{h^3} \sum_n \int\limits_{\text{BZ}} \boldsymbol{v} \frac{\omega}{\omega - \boldsymbol{v} \cdot \boldsymbol{k}} \boldsymbol{B} \cdot \boldsymbol{m} \frac{dn_F(\epsilon)}{d\epsilon} d\boldsymbol{p}, \tag{2}$$

where $\boldsymbol{v} = \boldsymbol{v}_n(\boldsymbol{p})$, $\boldsymbol{m} = \boldsymbol{m}_n(\boldsymbol{p})$, and $\epsilon = \epsilon_n(\boldsymbol{p})$ are, respectively, the velocity, the magnetic moment, and the energy of a Bloch state with quasimomentum $\boldsymbol{p}$ in the $n$th energy band [32] (the integral is taken over the Brillouin zone (BZ) and the sum is taken over all energy bands), $n_F$ is the Fermi-Dirac distribution function, $\omega$ and $\boldsymbol{k}$ are the frequency and the wavevector of the magnetic field, respectively.

It is tempting to extend the $\omega \to 0$ limit of Eq. (2) to a finite-size system, by simply replacing $k$ with $\sim 2\pi/L$, where $L$ is the thickness of the sample. However, such a replacement leads to $\boldsymbol{j} = \boldsymbol{0}$, contrary to the result that was found in [1]. In order to understand the reason for the discrepancy, we focus in more detail on the actual meaning of the $\omega \to 0$ limit in the bulk theory in case of strongly inhomogeneous magnetic field.

To be specific, we consider a sample having the geometry of a slab of thickness $L_\perp$, where the magnetic field $\boldsymbol{B} \parallel \boldsymbol{e}_z$ is localized homogeneously within the plane $x = 0$, and we measure the current density at some other point $x$. (See Fig. 1a.) If the slab is infinitely thick, $L_\perp \to \infty$, then by going to the real space in Eq. (2), one gets $j^z(x) = K(x|\omega)B^z$. It can be easily shown that the response function $K$ decays as $K(x|\omega) \sim \exp(i\omega x/v_{\text{F}})/\omega x^2$ at large distances, $x \gg v_{\text{F}}/\omega$, while it reaches some constant value at $x \lesssim v_{\text{F}}/\omega$. What it means is that the current is localized within the region $|x| \lesssim v_{\text{F}}/\omega$, which grows with decreasing $\omega$ to zero, while the integral of $j^z$ over this region is not sensitive to $\omega$. Such non-locality of a response function is typical of ballistic systems, and the lengthscale $v_{\text{F}}/\omega$ corresponds to typical distances that the electrons travel during one oscillation period $2\pi/\omega$.

If we now consider the case of finite $L_\perp$, such that $\omega \lesssim v_{\text{F}}/L_\perp$, the boundaries cannot be ignored anymore. Whatever the profile of the magnetic field $B^z(x)$ is, electrons that have been

---

disbalance in chemical potentials, $\mu_{\text{R}} \neq \mu_{\text{L}}$, the former effect was called differently in the literature, either the Gyrotropic Magnetic Effect (GME) [31] or the dynamic CME [29]. However, we prefer to stick to the original name "Chiral Magnetic Effect" for the effect that is studied in this paper.

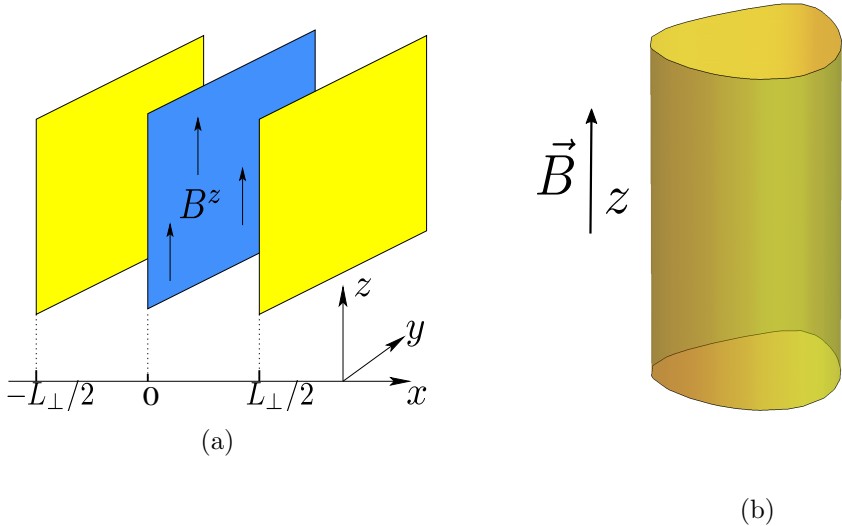

Figure 1: **(a)** Sample of a Weyl semimetal having the geometry of a slab, which is used to illustrate the non-local nature of the adiabatic CME. The material is located in the region $-L_\perp/2 < x < L_\perp/2$, and is infinite in both $y$ and $z$ directions. The oscillating magnetic field is localized within the infinitely thin plane $x = 0$. **(b)** Sample having the geometry of a cylinder. The $z$-axis is chosen along the cylinder's axis, and the cylinder is infinite in this direction. The magnetic field (both oscillating and the static components) is homogeneous.

affected by this field will have enough time to reach a surface and bounce back into the bulk. With decreasing frequency the number of reflections per cycle for each individual particle will increase, making the contribution of the physics of scattering at the boundary increasingly important. Even in the simple case when the boundaries are translationally invariant along $z$, so that the $z$-component of an electron's momentum, $p^z$, is conserved upon reflection, the effect may be quite non-trivial. Indeed, if the Weyl quasiparticles have an anisotropic velocity tensor, conservation of $p^z$ *does not necessarily* imply conservation of $v^z$ (the $z$-component of the velocity vector), which means that the reflection event may change the contribution of an electron to the current $j_z$. This semiclassical effect of the boundary is further complicated by the presence of the topologically protected surface states known as the Fermi arcs [15, 34, 35].

In order to incorporate all the boundary effects into one unified framework we proceed to the analysis of the bounded system within the quantum linear-response theory. We focus on a purely ballistic situation neglecting any effects of disorder and interparticle scattering. In our analysis we assume that the sample has the shape of a cylinder with an arbitrary base and that one of the main crystal axes of the material coincides with the cylinder's axis, which we denote as $z$. (See Fig. 1b.) The sample is placed in a homogeneous AC magnetic field also pointing in the direction of the cylinder's axis

$$B = B_{\text{AC}} \cos \omega t + B_0, \tag{3}$$

where we have also included a constant background field $B_0$. Rather than looking into the inhomogeneous and presumably complicated distribution of the current density $\boldsymbol{j}$, we focus

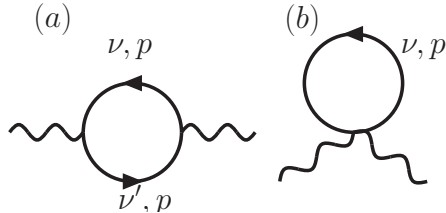

Figure 2: Feynman diagrams describing the linear response of the current $I$ to the oscillating magnetic field $B_{\mathrm{AC}}(t) \propto \cos \omega t$ in the first-principle Kubo approach. The wavy lines correspond to the electromagnetic vector potential, lines with arrows – to electron propagators. $p$ denotes the $z$-momentum of a virtual electron, $\nu$ ($\nu'$) is an additional discrete index that characterizes this state. (a): the contribution $I^{(a)}$ that describes the transitions between different energy levels ($\nu \neq \nu'$); (b): the contribution $I^{(b)}$ that does not involve the transitions.

on the total linear response current

$$I = S_\perp \frac{e^2}{h^2 c} \varkappa B_{\mathrm{AC}} \cos \omega t, \tag{4}$$

where $S_\perp$ is the cross-sectional area of the cylinder in the plane orthogonal to its axis.

## 2 Generalized Landauer-Büttiker formula for the adiabatic CME

We perform the linear response calculation for a system which is prepared in thermal equilibrium at temperature $T$. In the absence of driving, every eigenstate of a single particle Hamiltonian (or, simply, an orbital) is characterised by two quantum numbers, the projection of the Bloch momentum on the cylinder axis, $p$, and a label, $\nu$, that enumerates energy levels at given $p$ in the order of increasing energy. We denote by $f_\nu(p)$ the equilibrium occupation number of a given orbital.

The non-equilibrium current in the linear response theory is given by the Kubo formula [36] in which one can identify two different contributions $I = I^{(a)} + I^{(b)}$. The two Feynman diagrams for the polarization tensor giving rise to these different contributions are shown in Fig. 2. The diagram (a) describes the effect of the change in the single-particle density matrix due to the application of the AC perturbation. The corresponding contribution to the current is

$$I^{(a)}(\omega) = -\frac{eB_{\mathrm{AC}}}{h} \sum_{\nu,\nu'} \int_{\mathrm{BZ}} dp \; v^z_{\nu\nu'}(p) M^z_{\nu'\nu}(p) \frac{f - f'}{\epsilon + \hbar\omega - \epsilon'}. \tag{5}$$

Here $p$ runs here over the one-dimensional Brillouin zone (BZ) of the cylinder, $\epsilon$ ($\epsilon'$) is the energy of the state with quantum numbers $p$ and $\nu$ ($\nu'$), $f = f_\nu(p)$, $f' = f_{\nu'}(p)$. We have introduced the notation $A_{\nu\nu'}(p)$ for matrix elements of an operator $\hat{A}$ that is diagonal with respect to the quasimomentum, $\langle \nu, p | \hat{A} | \nu', p' \rangle = h\delta(p - p')A_{\nu\nu'}(p)$, $\hat{v}^z$ is the component of the velocity operator along the $z$-direction, $\hat{v}^z = i[\hat{H}, z]$, $\hat{M}^z$ is the magnetic moment along the same direction, $\hat{M}^z = -\partial_B \hat{H}$. Both operators $\hat{v}^z$ and $\hat{M}^z$ are diagonal with respect to the quasimomentum $p$ due to the translational invariance along the cylinder axis [37].

The second contribution to the Kubo formula (the diagram (b) in Fig. 2), $I^{(b)}$, comes from the non-trivial change of the current operator as resulting from the change in the magnetic field. Indeed, the current is proportional to the average velocity of the electrons, and the velocity in its turn can be sensitive to the magnetic field. The algebraic expression for such a contribution is

$$I^{(b)} = \frac{eB_{\text{AC}}}{h} \sum_\nu \int\limits_{\text{BZ}} dp f \left(\frac{\partial \hat{v}^z}{\partial B}\right)_{\nu\nu}, \tag{6}$$

where the operator $\partial_B \hat{v}^z \equiv i[\partial_B \hat{H}, z]$ is diagonal with respect to $p$. By going to the static limit $\omega = 0$, and by repeating the manipulations similar to [17, 31] (see Appendix A), we find

$$I(\omega = 0) = \frac{eB_{\text{AC}}}{h} \sum_\nu \int\limits_{\text{BZ}} dp f_\nu(p) \frac{\partial^2 \epsilon_\nu(p)}{\partial B \partial p}. \tag{7}$$

This formula is a finite-temperature generalization of Eq. (6) of [1]. It has a simple intuitive meaning, which is also how it was introduced in [1]. Indeed, if the frequency of the driving is vanishingly small, the occupation number of any given orbital is conserved by virtue of the adiabatic theorem. At the same time, the expectation value of the velocity of an electron occupying the orbital changes as

$$\delta v_\nu^z = \frac{\partial^2 \epsilon_\nu(p)}{\partial p \partial B} \delta B.$$

Note that formula (7) is a significant simplification of the Kubo formula, because it reduces the calculation of the current to the analysis of the single-particle *energy spectrum* only. Some further discussion of the zero-temperature limit of the formula (7) and its relation to the Landauer-Büttiker formula in mesoscopic physics can be found in [1].

If one integrates the expression (7) by parts with respect to $p$, the new integrand is $\partial_p f \partial_B \epsilon$. At zero temperature, the derivative $\partial_p f$ is a delta-function peak centered at the Fermi energy $\epsilon = \varepsilon_{\text{F}}$, which means that only the orbitals at the Fermi level contribute to the current. For non-zero temperature, the derivative $\partial_p f$ is a broadened peak with support $|\epsilon - \varepsilon_{\text{F}}| \lesssim T$, which means that still only the orbitals with energies close to the Fermi level contribute, at small enough temperatures. As a consequence, there is *no CME* in gapped systems (i.e. in systems with vanishing density of states at $\epsilon = \epsilon_F$). Another conclusion is that CME in the low-temperature limit can be described using effective low-energy theory alone.

Applicability conditions for formula (7) follow directly from its derivation. Firstly, in order to transform equations (6) and (5) into Eq. (7) one has to assume that the driving frequency $\omega$ is much less than the spacing between any pair of levels associated with a non-vanishing matrix element $M_{\nu\nu'}$. Secondly, our derivation only works in the *ballistic* regime that is if the single-particle scattering rate is much less than $\omega$. This, in turn, implies that the scattering rate needs to be much less than the level spacing at any given value of $p$.

There are at least two ways to make the inter-level spacing large enough, in order to ensure the adiabatic evolution. One way is to make the cylinder small enough in one of the transversal dimensions, which leads to stronger finite-size quantization of the energy levels. The other way is to apply strong magnetic field, which will lead to Landau quantization.

# 3 Example: adiabatic CME in a quantizing background magnetic field

Next, we apply Eq. (7) to the largely unexplored case of the chiral magnetic effect in the presence of a strong uniform background magnetic field $B_0$, focusing on the case of vanishing temperature $T = 0$. We show how formula (7) can be used to separate the contributions of the bulk and the surface states to the total current, $I = I_{\text{bulk}} + I_{\text{surf}}$. We demonstrate that in the approximation neglecting any gradient corrections to the Weyl spectrum, the bulk current vanishes and the chiral magnetic effect becomes a purely surface phenomenon. We also derive a simple formula for the surface contribution.

For simplicity, we consider the cylinder having circular cross section with radius $R$, so the boundary is described by the equation $x^2 + y^2 = R^2$. In such case, orbitals are classified by $(p, m, n)$, where the original index $\nu$ comprises the integer-valued angular momentum quantum number $m$ and the rest, $n$. Moreover, we will consider magnetic length $l_{\text{B}} = \sqrt{\hbar c / e B_0}$ that is much smaller than the radius, $l_{\text{B}} \ll R$, which is the quantitative measure of how strong the field $B_0$ is.

In an infinite sample all energy levels would be degenerate in $m$, so for the bulk states $\epsilon_\nu = \epsilon_n(p)$. In contrast, the orbitals localized close to the cylinder's surface (surface states) have a non-trivial dependence of the energy on both $p$ and $m$, $\epsilon = \epsilon_n(p, m)$. For these orbitals, angular momentum $m$ can be interpreted in terms of the momentum in the angular direction $p_{\parallel} = \hbar m / R$ (transversal momentum), and in the considered limit of large $R$ the variable $p_{\parallel}$ can be effectively regarded as a continuous one.

The bulk contribution to the current results from replacing the energies by their bulk expressions $\epsilon_n(p)$, and by taking into account that the number of orbitals per unit cross-section area (for each given Landau level $n$) is $e B_0 / hc$, and by considering only the states below the Fermi level $\varepsilon_{\text{F}}$,

$$I_{\text{bulk}} = \frac{e^2 B_{\text{AC}} B_0}{h^2 c} S_\perp \sum_n \sum_{p_F} \frac{\partial \epsilon_n(p_F)}{\partial B} \text{Sgn}\, v_n^z(p_F). \tag{8}$$

Here $S_\perp = \pi R^2$ is the cross section of the cylinder, and the sum goes over all solutions $p_F$ of the equation $\epsilon_n(p_F) = \varepsilon_{\text{F}}$.

The contribution $I_{\text{surf}}$ of the surface states has the form

$$I_{\text{surf}} = \frac{e^2 B_{\text{AC}}}{h^2 c} S_\perp \sum_n \int_{\text{BZ}} dp \left( v_n^z(p) \text{Sgn}\, \partial_{p_{\parallel}} \epsilon_n \right) \Big|_{\epsilon_n = \varepsilon_{\text{F}}} \rho_n(p), \tag{9}$$

and the direct derivation from Eq. (7) is given in Appendix A. Here $\rho_n(p)$ is 1 whenever $\epsilon_n(p, p_{\parallel}) = \varepsilon_{\text{F}}$ has a solution (for given $p$ and $n$) and 0 otherwise. Eq. (9) can be understood from the bulk-boundary correspondence as follows. Using the conservation of the momentum along $z$, one can perform the dimensional reduction from 3 to 2 dimensions. The resulting 2D lattice Hamiltonian can be characterized by a discrete Chern number, which is a piecewise constant of $p$. We recall that for a Weyl semimetal in absence of magnetic field the jumps of this function appear whenever $p^z = p$ plane contains a Weyl node [15, 34, 35]. (For realistic magnetic fields, when $l_{\text{B}}$ is much larger than the lattice spacing, the positions of the discontinuities will be slightly shifted, but the overall picture does not change.) For values of $p$ that have non-vanishing Chern number $c_N \in \mathbb{Z}$, there exist edge states which implies $\rho_n(p) = 1$.

At the same time, the bulk is gapped and experiences the quantum Hall effect with the Hall conductance $\sigma_H = e^2 c_N/h$. In presence of a slowly-varying magnetic field, the quantum Hall effect will ensure the adiabatic inflow of charge at the boundary.[3] Indeed, by Faraday's law, $\sigma_H \dot{\boldsymbol{B}} = -\boldsymbol{\nabla} \times (\hat{\sigma}_H \boldsymbol{E}) = -\boldsymbol{\nabla}\boldsymbol{j}$. Using the continuity equation, the amount of the charge that is pushed from the bulk to the boundary is $dQ = \sigma_H S_\perp dB$. The contribution of this surface charge to the current will be $v_n^z dQ$, which leads to Eq. (9) after the integration over $p$.

We see that $I_{\text{bulk}}$ and $I_{\text{surf}}$ scale in the same way with the cross-section, as it was discovered in [1].

In order to further illustrate the importance of the surface contribution, we apply the formula (8) to a Weyl semimetal to demonstrate that the *bulk* CME current is vanishing (in a sufficiently strong background magnetic field $B_0$). The minimal effective theory around a Weyl node is described by the effective Hamiltonian

$$\mathcal{H}(\boldsymbol{p}) = \chi v_{\text{F}} \boldsymbol{p} \cdot \boldsymbol{\sigma}, \tag{10}$$

where $\chi = \pm 1$ describes the chirality of the fermion (right for the upper sign, left – for the lower sign), $v_{\text{F}} > 0$ is the quasiparticle velocity,[4] and we assume for simplicity that the Weyl node is located at zero energy and at $\boldsymbol{p} = \boldsymbol{0}$. The dispersion relation of the bulk Landau levels is given by [38, 39]

$$\varepsilon_n^{\text{LL}} = \begin{cases} -\chi v_{\text{F}} p, & (n = 0) \\ \text{Sgn}\, n \cdot v_{\text{F}} \sqrt{p^2 + 2\frac{|enB_0|\hbar}{c}}. & (n \neq 0) \end{cases} \tag{11}$$

Since the energy of the $n = 0$ level does not depend on magnetic field whatsoever, according to Eq. (8) this particular level does not contribute to the current $I_{\text{bulk}}$. While the energies of the $n \neq 0$ levels do involve the magnetic field, these energies are *even* with respect to $p$, so their contribution to the integral (8) vanishes as well. The only caveat is non-minimal corrections to the effective Hamiltonian, which we discuss in more detail in the accompanying paper [40]. (Conclusion is that the corrections are suppressed as the ratio of the distance from the Weyl node to $\epsilon_F$ and the bandwidth of the Bloch band.)

## 4 Discussion

In order to measure the CME in an experiment, one can attach two ends of a ballistic WSM wire to Ohmic contacts made of a non-magnetic material with inversion symmetry. (This way, we make sure that the effect comes from the wire alone.) In order to measure the coefficient $\varkappa$, the contacts need to be good, which in practice means that in the quantizing magnetic field the measured two-terminal conductance of the wire has to be $(e^2/h)S_\perp/2\pi l_{\text{B}}^2$.

For the existing WSM crystals (e.g. TaAs used in the experimental work [41]), which have the transport lifetime of quasiparticles $\tau_{\text{tr}} \sim 10^{-11}$ sec, which implies that the measurements should be done at frequencies $\omega \gtrsim 2\pi/\tau_{\text{tr}} \sim$ THz in order to have ballistic behaviour.[5] By

---

[3]Note that the direction of the charge flow depends on the sign of $c_N$, and on the other hand this direction is given by the sign of $\partial\epsilon_n/\partial p_\parallel$ (see Appendix A for more details), so $c_N = \text{Sgn}\, \partial\epsilon_n/\partial p_\parallel$.

[4]We consider here isotropic effective theory, meaning that the velocity is the same in all possible directions of motion. However, in the anisotropic case, $\mathcal{H}(\boldsymbol{p}) = \sum_{i,j} p^i v^{ij} \sigma^j$ (where the eigenvalues of $v^{ij}$ are different) the bulk current vanishes as well.

[5]Note however that this value of $\tau_{\text{tr}}$ was found in the absence of a magnetic field, while in strong magnetic field it can be quite different.

taking $v_{\mathrm{F}} \sim 10^7$ cm/ sec from [42] (ab initio calculations for TaAs), we find the upper bound on the radius of the wire: $L_\perp \lesssim 2\pi v_{\mathrm{F}}/\omega \lesssim v_{\mathrm{F}}\tau_{\mathrm{tr}} \sim 1\mu$m.

Finally we would like to note that the CME can be present in systems that are not Weyl semimetals in the sense that they are described by an effective Hamiltonian other than (10). The important ingridients to have the effect in the bulk is the broken $P$-symmetry [31] (note that $\hat{\varkappa}$ in Eq. (1) is a pseudoscalar) and finite density of states at the Fermi level. In order to have the surface current, we need the bulk that is characterized by a non-trivial Chern number. The candidate materials include double-Weyl semimetals (like $SrSi_2$ [43]), type-II WSMs (like $WTe_2$ [44]) and materials having nexus fermions in their spectrum (like WC [45]).

## Acknowledgements

The authors are grateful to Paul Baireuther, Carlo Beenakker, Igor Gornyi, Alexander Mirlin, and Laurens Molenkamp for fruitful discussions and useful comments. The research was supported by the Delta Institute for Theoretical Physics.

## A   Appendix

In this Appendix, we provide the intermediate calculations that are required in order to get Eq. (7) from Eqs. (5) and (6) in the $\omega \to 0$ limit. Then we show how to extract the contribution of the surface states from Eq. (7) to get Eq. (9).

The calculations that are required to get Eq. (7) are similar to the steps taken in [17,31], but one should notice, however, the fundamental difference: in [17,31] the authors consider Bloch crystals that are inifinite in all *three* directions, while here we consider crystals that have shape of an infinite cylinder. It means that in the present analysis, the translational invariance is broken in the two directions that are orthogonal to the cylinder's axis.

By differentiating both the normalization condition $\langle \nu, p|\nu', p'\rangle = h\delta_{\nu\nu'}\delta(p - p')$ and the Schrödinger equation $(\hat{H} - \epsilon_\nu)|\nu, p\rangle = 0$ with respect to $B$ ($\hat{H}$ is the single-particle Hamiltonian), we get $(\partial_B\langle \nu, p|)|\nu', p'\rangle = -\langle \nu, p|\partial_B|j, \boldsymbol{p}'\rangle$ and $\langle \nu', p'|\hat{M}^z|\nu, p\rangle = (\epsilon_{\nu'} - \epsilon_\nu)\langle \nu', p'|\partial_B|\nu, p\rangle - \delta_{\nu\nu'}h\delta(p - p')\partial_B\epsilon_\nu$. By using the resolution of the identity operator $\sum_{\nu'}\int_{\mathrm{BZ}} dp'|\nu', p'\rangle\langle \nu', p'| = \not\Vdash$, we find

$$I^{(a)}(\omega = 0) = \frac{eB_{\mathrm{AC}}}{h^2}\sum_\nu \int\limits_{\mathrm{BZ}} dp\,dp'\, f_\nu \left[\langle \nu, p|\hat{v}^z\partial_B|\nu, p'\rangle + (\partial_B\langle \nu, p|)\hat{v}^z|\nu, p'\rangle\right]. \qquad (12)$$

Noting that $I^{(b)}$ can be re-written as

$$I^{(b)} = \frac{eB_{\mathrm{AC}}}{h^2}\sum_\nu \int\limits_{\mathrm{BZ}} dp\,dp'\, f_\nu \langle \nu, p|\frac{\partial\hat{v}^z}{\partial B}|\nu, p'\rangle, \qquad (13)$$

we arrive at Eq. (7).

Now we will show how to extract the contributions of the surface states from Eq. (7). It is known that the radius $r$ around which the orbital with given $m$ and $n$ is localized, is given by $r = \sqrt{2\hbar cm/eB}$ [46]. Thus, whenever we change the magnetic field, the energy of a surface state changes mainly due to the shift of its spatial position, which can be effectively taken

into account by the replacement $\partial_B\epsilon \to (eR^2/2\hbar c)\partial\epsilon/\partial m$. Next, we replace summation over $m$ in Eq. (7) by integration over $p_\parallel = \hbar m/R$, which is a valid procedure in our case of $R \gg l_B$. By taking the resulting integral by parts, we arrive at Eq. (9).

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
