# Peer review of "Adiabatic Chiral Magnetic Effect in Weyl semimetal wires"

_SciPost Physics_

## Round 1 · Referee Report · Anonymous (Referee 1) · 2018-2-1

Strengths

  • The paper addresses a subtle effect that has not yet been directly observed in Weyl semimetals. Aiming at a clear picture of this effect in practical conditions is clearly desirable.

Weaknesses

  • See below.

Report

In this work, the authors study the Chiral Magnetic Effect (CME), the generation of a current by an applied magnetic field, in Weyl semimetals in finite samples. Originally proposed in the context of relativistic plasmas, the observability of the CME has recently been debated in many works as a potential smoking gun signature of Weyl semimetals. These works have clarified that the CME response is only observable either with an unequal, non-equilibrium population of nodes of opposite chirality, or when the magnetic field is at finite frequency. The limit q=0, w->0 with w>>q is the one studied in this work and called adiabatic CME, and has been called gyrotropic magnetic effect or DC limit of the dynamical CME in previous works. The question asked here is whether this adiabatic CME is modified when finite samples are considered. Below I review the current status and their claims in more detail, and argue that their results are not sufficiently novel or distinct from previous works, and in particular their proposal of a CME completely determined by boundary effects is inconsistent with a previous result. I do not believe the paper is suitable for publication in SciPost.

A Kubo-type formula is used to obtain Eq. 7, the main result of this work. This is then used to compute the mentioned CME with a low energy model for a Weyl semimetal, in the presence of a background field B_0 in addition to the perturbing field B(w). They authors call this a “largely unexplored case”, but it is in fact addressed in Refs. 1 and 18. Ref 1 explicitly considers a lattice model of a Weyl semimetal, and uses a Landauer-Buttiker approach to compute this CME both at zero and finite background field. At zero field, the result is that the surface states contribute twice as much as the bulk, so the effect is indeed modified by the presence of a surface. At finite background field (see their Fig. 5), the total effect remains unchanged. But surface and bulk contribute but the effects cannot be clearly separated as clearly. This is at odds with the claim made in this work that the bulk does not contribute. Moreover, the main equation used to derive these results was already derived in Ref. 1. I do not understand in what sense this derivation was “heuristic” as the authors say.

A similar equation was also derived in Ref. 18 (their Eq. 8), and used to address the adiabatic CME of a model without Weyl nodes or Berry curvature (what matters for this effect is the orbital magnetic moment which requires neither) with open boundaries. The results are also for finite magnetic field, and are also finite, reiterating a previous result that Weyl nodes are not required for CME.

My opinion is that neither Eq. 7 nor considering the case of a finite B_0 represent a significant advancement over previous works, and that Ref 1 is quite explicit and contradicts this work. Even if the discrepancies were clarified and the work of Refs. 1 and 18 were properly acknowledged, I still think that the overall importance and novelty of this work would not be enough to merit publication in SciPost.

Requested changes

See above.

---

## Round 1 · Referee Report · Anonymous (Referee 2) · 2018-2-18

Strengths

  1. Timely paper.
  2. Adresses an open question posed by a previous prominant publication.

Weaknesses

  1. Motivation not clearly explained.
  2. Not put in the right context, the introduction is lacking and the discussion isn't sufficient.
  3. The uniqueness of the analysis to the claimed phases discussed is questionable.

Report

I reviewed the manuscript “Adiabatic Chiral Magnetic Effect in Weyl semimetal wires” by Ivashko and Cheianov. The paper discusses a CME that appears in equilibrium due to an oscillating magnetic field. This work follows up on a recent paper by the group of Beenakker (Ref. [1]), where a scattering theory was formulated within the Landauer Buttiker approach for Weyl semimetal nanowires. Ref. [1] posed a couple of intriguing puzzles concerning the role of Fermi arcs in transport, showing that they carry a non-negligible contribution even in the limit of a very large system. The authors of the current manuscript attempt to clarify the origins of that contribution, deriving it directly from the Kubo formula.

The topic of the paper is certainly timely, and providing a novel perspective is valuable since Ref. [1] did int leave some room for additional analysis and further support for some of their results. In that respect, this paper should be published, but I do have reservations regarding some subtle points that I feel are not properly address and clarified in the current version.

At this point, I believe that the most puzzling aspect of this manuscript is the discussion preceding the calculation presented in the paper. In particular, I think the authors should clarify substantially what is the role of the Berry curvature monopole structure vs. the role of the orbital moment in their theory. Already at the early stage of the paper, the authors cite the current that they calculate, in equation 2, and this current is directly proportional to the orbital magnetic moment m. This piece of the current is just one piece contributed by the orbital magnetization and goes to zero when that moment is zero. This piece is also not unique to Weyl semimetals - it does not require monopoles to exist in the band structure or Fermi arcs. Any system that posses an orbital moment and surface states can in principle have this current be non zero, and so I am not entirely sure why the preceding discussion stresses the topological nature of Weyl semimetals. In short: this piece is the non-topological piece of the current, as explained by Ma and Pesin, and so is not unique to Weyl semimetals.

There is some discussion related to this in footnote 2, which I think should be presented and extended in the body of the paper, and another mentioning of this in the concluding remark, but in my opinion, the distinction between the CME and what is sometimes called GME should be discussed much more thoroughly. Is it the author’s claim that the orbital moment of the arcs due to their circular motion the cause for the surface current? This, for example, is not clear from the current discussion.

Requested changes

  1. Extend the introduction of the paper.
  2. Discuss more thoroughly the origins of equation 2 and any additions that are not considered.
  3. Discuss the uniqueness for Weyl semimetals and topological structure if applicable, if not, explain why.

---

## Editorial Decision

awaiting_resubmission